# Prevalence of migraine in adults with celiac disease: A case control cross-sectional study

**Mohammad M. Fanaeian** [1], **Nazanin Alibeik** [2]*, **Azita Ganji**[3], **Hafez Fakheri** [4], **Golnaz Ekhlasi**[5], **Bijan Shahbazkhani**[6]

**1** Division of Gastroenterology and Liver Diseases, Imam Khomeini Hospital Complex, Tehran University of Medical Sciences, Tehran, Iran, **2** Clinical Research Development Center, Firoozgar Hospital, Iran University of Medical Sciences, Tehran, Iran, **3** Department of Gastroenterology and Hepatology, Mashhad University of Medical Sciences, Mashahd, Iran, **4** Gut and Liver Research Center, Non-communicable Disease InstitueMazandaran University of Medical Sciences, Sari, Iran, **5** Golnaz Ekhlasi; Digestive Disease Research Institute, Shariati Hospital, Tehran University of Medical Sciences, Tehran, Iran, **6** Department of Gastroenterology and Liver Diseases, Tehran University of Medical Sciences, Tehran, Iran

* Alibeik.n@iums.ac.ir

## Abstract

### Aim

Celiac disease (CD) is an immune-mediated disorder with various manifestations. The aim of this study was to evaluate the prevalence of gastrointestinal (GI) and extra-intestinal symptoms of celiac patients, especially migraine, and compare it with healthy individuals.

### Methods

We compared 1000 celiac subjects (CS) registered at our celiac center with the control group for headache-based on International Classification of Headache Disorders, third edition criteria and their GI symptoms. Besides, CS with migraine and non-migrainous headache were compared in terms of GI symptoms and accompanied conditions.

### Results

Headache was more common in CS than controls (34% vs 27% respectively, P value<0.001) and more prevalent in females (71.9% in females vs 28% in males, P value = 0.004). Moreover, the prevalence of migraine in CS was higher than controls (20.7 vs 11.9% respectively, P value<0.001). Furthermore, migraine was more prevalent in females with CD (80% in females vs 19% in males, P value = 0.033), and often without aura (76%). Abdominal pain (76.9%, P value = 0.025), diarrhea (54.9%, P value = 0.002), and constipation (42.9%, P value = 0.011) were the most common GI symptoms in CS with headache and more prevalent in CS with migraine. Conversely, type 1 diabetes mellitus was less common in CS with migraine than in CS with non-migrainous headache. (P value = 0.001). On multivariate logistic regression analysis, female sex (OR 1.50, 95%CI 1.22–1.83, P value < 0.001), and CD (OR 1.36, 95%CI 1.12–1.65, P value = 0.002) were independent predictors of headache, whereas age more than 60 years (OR 0.70, 95%CI 0.50–0.97, P value = 0.032) had a protective effect.

**Data Availability Statement:** All relevant data are within the manuscript and its Supporting information files.

**Funding:** The authors received no specific funding for this work.

**Competing interests:** The authors declared no potential conflicts of interest with respect to the research, authorship, and/or publication of this article.

## Conclusion

Headache especially migraine is more prevalent in CS than healthy controls. In addition, abdominal pain, diarrhea, and constipation are more common in CS with migraine than in CS with non-migrainous headaches. Therefore, evaluation of CD in patients with migraine and these simultaneous GI symptoms seems reasonable.

## Introduction

Celiac disease (CD) affects about 1% of the population worldwide [1]. It is an immune-mediated disorder occurs in genetically predisposed people after a gluten exposure [2]. This immune response subsequently causes intraepithelial lymphocytic infiltration, villous atrophy and crypt hyperplasia in intestinal mucosa [3]. CD can present with typical gastrointestinal (GI) symptoms such as bloating or diarrhea or with atypical extra-intestinal manifestations such as neurological disorders or it may be silent or asymptomatic [4]. There are reports of numerous CD patients in whom neurologic symptoms are the main complaint even in the absence of GI symptoms at the time of diagnosis [5,6]. Also, many individuals with neurologic disorders of unknown origin and migraine have been positive for anti-gliadin antibodies [7,8]. On the other hand, migraine is a throbbing often disabling unilateral headache generally associated with nausea, vomiting, photophobia and phonophobia [9–12]. About 6% of men and 17% of women have experienced migraine attacks in their lifetime and this severely affects their quality of life [13–15]. Although, the exact pathophysiology of migraine is not fully understood [16,17], the level of pro-inflammatory cytokines like tumor necrosis factor alpha (TNF α) and interleukin 1 Beta (IL 1β) have been increase in serum of patients during migraine attack [18]. Subsequently, these substances cause migraine attack by affecting the trigeminal nerve nociceptors. The relationship between migraine and GI disorders such as CD and Non celiac gluten sensitivity (NCGS) has been discussed in numerous studies [19–24], and there is a pro-inflammatory response in either CD after gluten exposure and migraine [25]. Moreover, some evidence show that migraine are dramatically improved by treatment of the underlying GI disease (i.e. Gluten- free diet in CD) [23,26,27].

On the other hand, some studies have reported an inverse relationship between migraine and type 1 diabetes mellitus [28,29], the pathophysiology of which is not exactly understood. However, vascular stiffness due to intimal thickening as well as a decrease in pain sensation following diabetic neuropathy and subsequent decrease in cerebrovascular reactivity may have protective effects for migraine [30,31]. Limited studies have been performed on the association of migraine with celiac disease in the Iranian population. we investigated the prevalence of migraine in CD patients for early diagnosis and appropriate treatment.

## Materials & methods

### Study population

In this case-control cross sectional multicentric study, we assessed the data of 1000 celiac patients over 18 years of age (mean 41.16 ± 15.12) from medical centers in different provinces of Iran (medical centers in Tehran, Khorasan Razavi and Mazandaran provinces), who were registered to celiac registry center at Shariati Hospital, Tehran, Iran between 2014 and 2019. Celiac disease was diagnosed based on its positive serologic markers including immunoglobulin A against tissue transglutaminase (Anti tTG IgA) and endomysial antibody (EMA), and

**Table 1. Demographic characteristics in celiac patients and control group.**

| | | Control group (n = 1000) | Celiac group (n = 1000) | P value |
|---|---|---|---|---|
| **Sex** | Female | 557 (55.7%) | 642 (64.2%) | <0.001 |
| | Male | 443(44.3%) | 358(35.8%) | |
| **Age** | 18–40 | 597(59.7%) | 569(56.9%) | 0.412 |
| | 41–60 | 294(29.4%) | 309(30.9%) | |
| | >60 | 109(10.9%) | 122(12.2%) | |

confirmed by endoscopy and duodenal biopsy (Marsh classification 3 and 4) in all subjects subsequently [32,33]. Celiac serological titers were assessed according to the ELISA method. We also used International classification of headache disorders 3rd edition (ICHD-3) criteria for headache diagnosis and classification [34]. The control group included 1000 healthy individuals over 18 years of age (mean 40.27±14.95) with negative celiac serologic markers and/or duodenal mucosal biopsy, who were recruited from hospital staff (physicians and nurses), friends, as well as patients family members. Subjects who had a history of inflammatory bowel diseases (IBD), previous head trauma, brain tumor, head and neck surgery or vascular problems were excluded from both groups. Out of 2000 participants, 642 (64.2%) and 557 (55.7%) were female from the celiac group and the control group, respectively (P value = 0.000). Characteristics of these two groups were shown in Table 1.

## Study protocol

For all participants, a checklist including general information, age, sex, medical conditions, symptoms at the time of diagnosis, which included GI and non-gastrointestinal symptoms, initial diagnostic tests (endoscopic examination, duodenal biopsy report) and treatment plan was filled at the first visit. Subjects who reported a history of headache at initial evaluation were further evaluated with a checklist based on ICHD-3 criteria for different types of primary headache (i.e., migraine without aura, migraine with aura and non-migrainous headaches). Finally, the data of the two groups were compared (Fig 1). Moreover, celiac subjects (CS) with

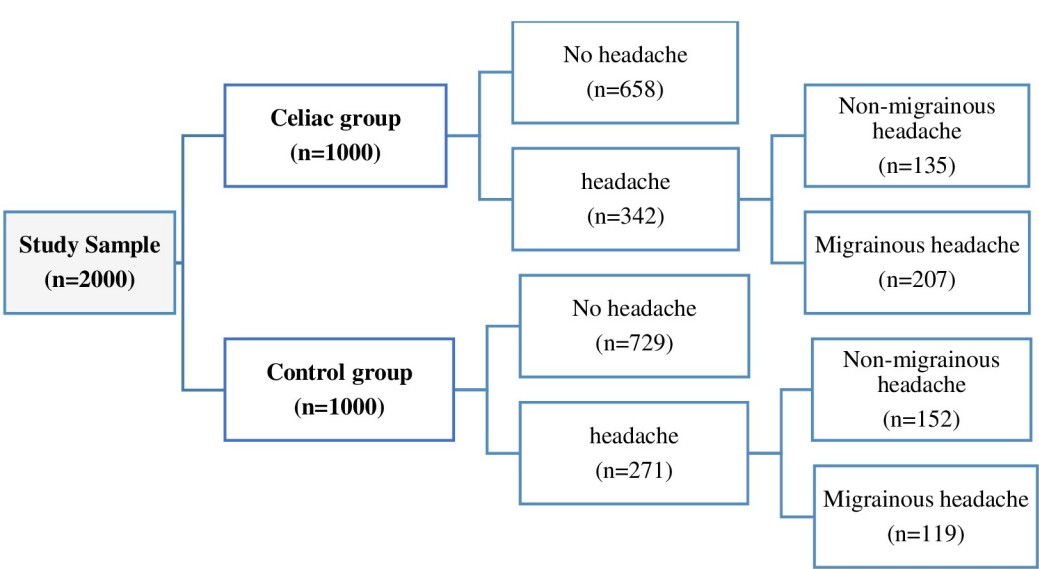

**Fig 1. Study nomogram.**

migraine and non-migrainous headache were compared in terms of GI symptoms and accompanied conditions.

## Ethics statement

The study was performed in accordance with appropriate guidelines and reviewed and approved by the Local Ethics Committee of Tehran University of Medical Sciences (Approval No: IR.TUMS.DDRI.REC.1396.10).

Address: Tehran University of Medical Sciences (TUMS), Poursina St., Qods St., Enqelab St., Tehran, Iran.

Tel No.: 0098–2164053419.

Fax No.: 0098–88962510.

Written consent was obtained from all participants after informing them.

## Statistical analysis

In this study, statistical analysis was done using SPSS version 16. The chi-square test and fisher's exact test were used to compare proportions. P values < 0.05 were considered statistically significant. A multivariate logistic regression analysis was done to evaluate the associations between having celiac disease, gender, age group and having headache.

## Results

342 (34%) celiac subjects (CS) and 271 (27%) controls reported headache, of which 246 (71.9%) of CS and 165 (60.9%) of controls were female. 207 (20%) CS and 119 (11%) subjects in control group had migraine. Also, 166 (80.2%) of celiac group and 83 (69.7%) of controls who reported migraine were female. Headache descriptions in CS and controls based on gender and age group were shown separately in Table 2, Figs 2 and 3. Forty-nine (23%) of CS and 22 (18%) of controls reported migraine with aura. The characteristics of headache and migraine in both groups were shown in Tables 3 and 4. We also found that abdominal pain (76.9%, P value = 0.025), diarrhea (54.9%, P value = 0.002), and constipation (42.9%, P value = 0.011) were the most common GI symptoms in CS with headache and were more prevalent in CS with migraine than those with non- migrainous headache. Conversely, type 1 diabetes mellitus was less common in CS with migraine than in CS with non-migrainous headache. (P value = 0.001). This comparison was shown completely in Table 5. On multivariate logistic regression analysis, female sex (odds ratio [OR] 1.50, 95% confidence interval [CI] 1.22–1.83, P value < 0.001), and celiac disease (OR 1.36, 95%CI 1.12–1.65, P value = 0.002) were independent predictors of headache; On the other hand, age more than 60 years (OR 0.70, 95%CI 0.50–0.97, P value = 0.032) had a protective effect (Table 6).

## Discussion

Our findings suggest that headache is more common in CS than controls (34% vs 27% respectively, P value< 0.001), And is more prevalent in females than males with celiac disease (71.9% in females vs 28% in males, P value = 0.004). In addition, the prevalence of migraine in CS is higher than controls (20.7 vs 11.9% respectively, P value<0.001). Furthermore, migraine was more prevalent in women than men with celiac disease (80% in women vs 19% in men, P value = 0.033), and was often without aura (76%). The migraine was often pulsatile (81.6%), unilateral (65%), and was accompanied by nausea and/or vomiting (67%), photophobia and phonophobia (60.9%). Dimitrova et al. [22] have also reported higher prevalence of headaches and migraine in celiac disease, gluten sensitivity, IBD (ulcerative colitis) than in the controls,

**Table 2. Headache descriptions in celiac subjects and controls based on gender and age group.**

| | | Controls | | | Celiac subjects | | | P-value |
|---|---|---|---|---|---|---|---|---|
| | | No headache (n = 729) | headache (n = 271) | P-value | No headache (n = 658) | headache (n = 342) | P-value | |
| Gender | Male | 337 (46.2%) | 106 (39.1%) | 0.044 | 262 (39.8%) | 96 (28.1%) | <0.001 | 0.004 |
| | Female | 392 (53.8%) | 165 (60.9%) | | 396 (60.2%) | 246 (71.9%) | | |
| Age | 18–40 | 430 (59%) | 167 (61.6%) | 0.227 | 377 (57.3%) | 192 (56.1%) | 0.097 | 0.387 |
| | 41–60 | 212 (29.1%) | 82 (30.3%) | | 192 (29.2%) | 117 (34.2%) | | |
| | >60 | 87 (11.9%) | 22 (8.1%) | | 89 (13.5%) | 33 (9.6%) | | |

in the United states. Gluten-sensitive group had the highest rate of headache (56%), to compare with celiac disease (30%), IBD (22%) and (14%) controls. Although, the prevalence of migraine diagnosed by ID Migraine criteria was 21% in celiac group, 40% in gluten sensitivity and 6% in control group. In our study compared to Dimitrova et al, [22] the celiac and control groups were more matched in age and sex. Furthermore, Dimitrova et al. found that the age more 65 years is a protective factor for migraine, whereas our study suggests the age more than 60 years is protective. Cicarelli et al. [35] and Briani et al. [5], reported a migraine prevalence of 32% and 5.6% in celiac patients in Italy, respectively. Although the sample size of our study was much larger than the studies mentioned, with different percent, they all indicate an increased prevalence of migraine in celiac disease. In addition, we assessed further characteristics of migraine (i.e., aura, pulsatile, nausea, vomiting, photophobia and phonophobia) in CD patients. We revealed abdominal pain (76.9%, P value = 0.025), diarrhea (54.9%, P

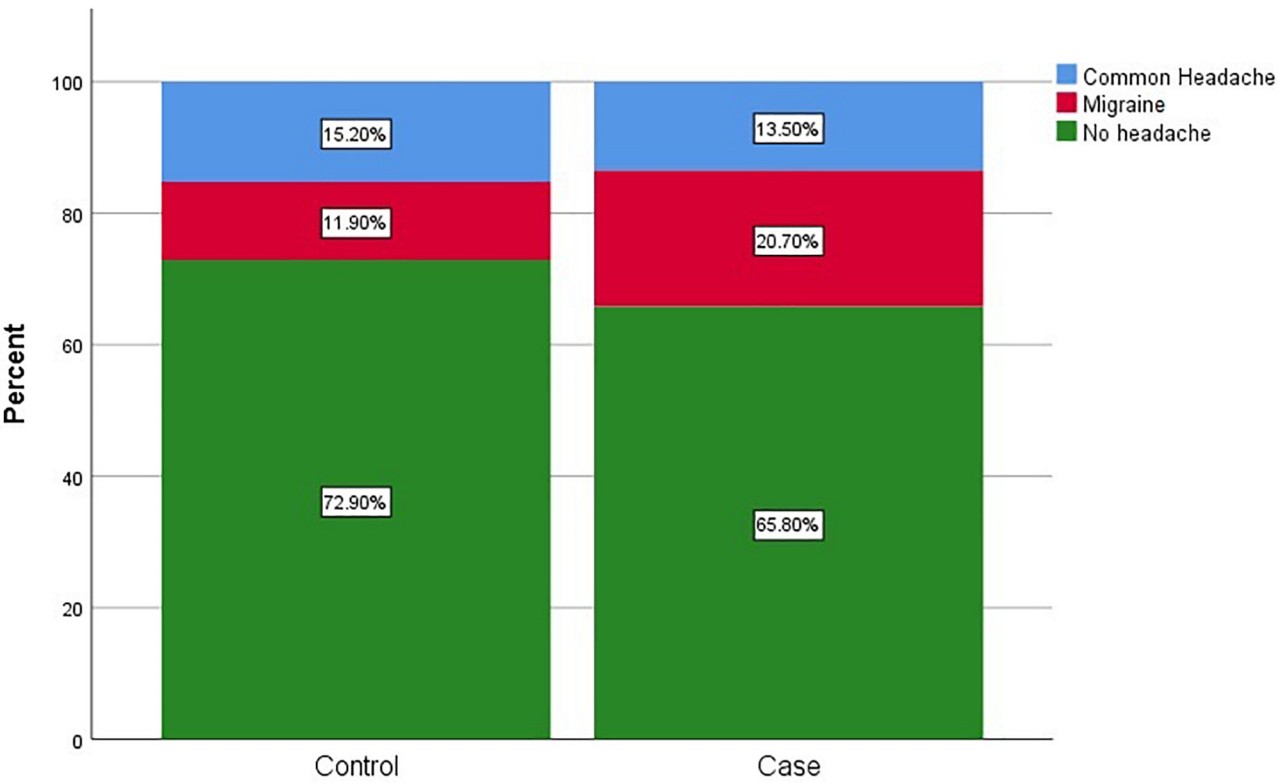

**Fig 2. Prevalence of different types of headaches in both celiac and control groups.**

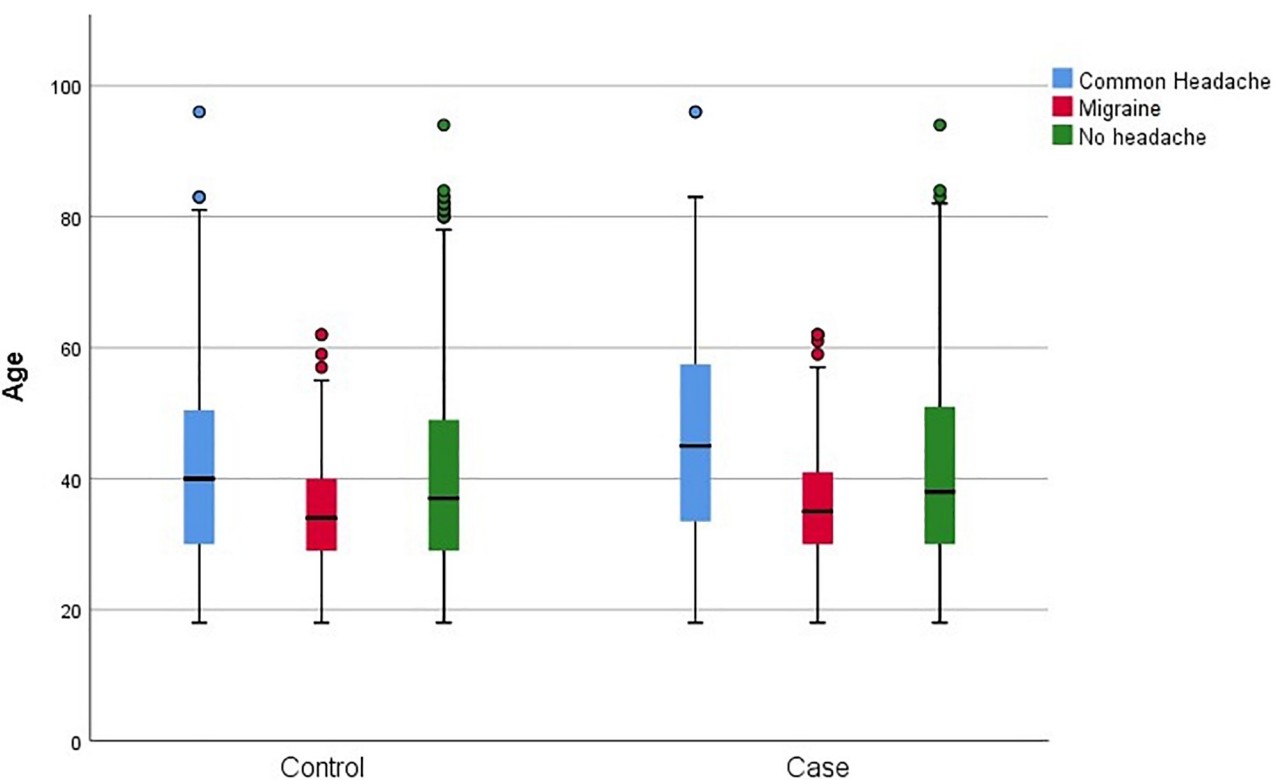

**Fig 3. Comparison of headache prevalence in different age groups between control and celiac subjects.**

value = 0.002), and constipation (42.9%, P value = 0.011) were the most common GI symptoms in CD patients with headache and were more prevalent in CD patients with migraine than those with non-migrainous headache. Although, according to ICHD-3 criteria, migraine may be associated with some GI symptoms such as abdominal pain [34], concurrent GI disorders like CD should be ruled out before attributing these symptoms to migraine. Some studies show that the physiopathology of migraine and some functional gastrointestinal diseases are relatively similar [36]. It is important to note that, regardless of celiac disease, migraines can be associated with some GI symptoms that may be helpful in understanding their physiopathology. On the other hand, as Hagen K et al. have revealed the inverse relationship between type 1 diabetes mellitus and migraine in non- celiac individuals [32], we also found that type 1 diabetes mellitus is less common in celiac patients with migraine than in celiac patients with non-

**Table 3. Comparison of different types of headaches in CD group vs controls.**

| | | Controls | | | Celiac subjects | | | P-value |
|---|---|---|---|---|---|---|---|---|
| | | Non-migrainous headaches (n = 152) | Migraine (n = 119) | P-value | Non-migrainous headaches (n = 135) | Migraine (n = 207) | P-value | |
| Gender | Male | 70 (46.1%) | 36 (30.3%) | 0.005 | 55 (40.7%) | 41 (19.8%) | <0.001 | 0.033 |
| | Female | 82 (53.9%) | 83 (69.7%) | | 80 (59.3%) | 166 (80.2%) | | |
| Age | 18–40 | 77 (50.7%) | 90 (75.6%) | <0.001 | 49 (36.3%) | 143 (69.1%) | <0.001 | 0.449 |
| | 41–60 | 55 (36.2%) | 27 (22.7%) | | 57 (42.2%) | 60 (29%) | | |
| | >60 | 20 (13.2%) | 2 (1.7%) | | 29 (21.5%) | 4 (1.9%) | | |

**Table 4. Characteristics of migraine in CD group vs controls.**

| Migraine characteristics | Controls (n = 119) | Celiac group (n = 207) | P value |
|---|---|---|---|
| **Aura** | 22 (18.5%) | 49 (23.7%) | 0.330 |
| **Unilateral** | 80 (67.2%) | 135 (65.2%) | 0.808 |
| **Pulsatile** | 95 (79.8%) | 169 (81.6%) | 0.770 |
| **Nausea and/or Vomiting** | 83 (69.7%) | 139 (67.1%) | 0.711 |
| **Photophobia and phonophobia** | 70 (58.8%) | 126 (60.9%) | 0.726 |

migrainous headache. (P value = 0.001). We also performed a multivariate logistic regression analysis to evaluate independent association between celiac disease and presence of headache; and found that female sex (odds ratio [OR] 1.50, 95% confidence interval [CI] 1.22–1.83, P value < 0.001), and celiac disease (OR 1.36, 95%CI 1.12–1.65, P value = 0.002) were independent predictors of headache. On the other hand, age more than 60 years (OR 0.70, 95%CI 0.50–0.97, P value = 0.032) had a protective effect (Table 6).

Compared to Ruggieri et al. [37] we found that the prevalence of headache was higher in adults with celiac disease than in children. On the other hand, Shahbazkhani et al. have described the headache as one of the most common extra-intestinal symptoms in NCGS patients [20]. Perhaps we can conclude that some factors other than genetic susceptibility and gluten exposure are involved in the occurrence of headache in celiac patients.

The different sex distribution between the two groups (P = 0.000) and the lack of measurement of other serologic markers such as anti-neuronal antibodies [33] in subjects with

**Table 5. Evaluation of GI symptoms and accompanied conditions in CD subjects with migraine in comparison with CD subjects with non-migrainous headache.**

| Symptoms and associated Conditions | CD subjects with Non-migrainous Headaches (n = 135) | CD subjects with migraine (n = 207) | P-value | Total CD group (n = 342) |
|---|---|---|---|---|
| **Hypothyroidism** | 23 (17.03%) | 34 (16.4%) | 0.756 | 57 (16.6%) |
| **Hyperthyroidism** | 5 (3.7%) | 7 (3.3%) | 0.829 | 12 (3.5%) |
| **Type I Diabetes mellitus** | 12 (8.8%) | 3 (1.4%) | 0.001 | 15 (4.3%) |
| **Type II Diabetes mellitus** | 2 (1.4%) | 7 (3.3%) | 0.299 | 9 (2.6%) |
| **Abdominal Pain** | 97 (71.8%) | 166 (80.1%) | 0.025 | 263 (76.9%) |
| **Constipation** | 48 (35.5%) | 99 (47.8%) | 0.011 | 147 (42.9%) |
| **Diarrhea** | 62 (45.9%) | 126 (60.8%) | 0.002 | 188 (54.9%) |
| **Nausea** | 60 (44.4%) | 97 (46.8%) | 0.466 | 157 (45.9%) |
| **Vomiting** | 28 (20.7%) | 54 (26%) | 0.158 | 82 (23.9%) |
| **Flatulence** | 97 (71.8%) | 166 (80.1%) | 0.064 | 263 (76.9%) |
| **GERD** | 68 (50.3%) | 112 (54.1%) | 0.252 | 180 (52.6%) |

GERD = Gastro-esophageal reflux disease; GI = Gastro-intestinal; CD = Celiac disease.

**Table 6. Univariate and multivariate analysis for headache prediction.**

| | | Odds Ratio (CI95%) | | | |
|---|---|---|---|---|---|
| | | Univariate Analysis | P-value | Multivariate Analysis | P-value |
| **Celiac** | **Yes / No** | 1.40 (1.15–1.69) | 0.001 | 1.36 (1.12–1.65) | 0.002 |
| **Gender** | **Female / Male** | 1.54 (1.26–1.88) | <0.001 | 1.50 (1.22–1.83) | <0.001 |
| **Age** | **41–60 / 18–40** | 1.11 (0.89–1.37) | 0.343 | 1.09 (0.88–1.35) | 0.430 |
| | **>60 / 18–40** | 0.70 (0.51–0.97) | 0.034 | 0.70 (0.50–0.97) | 0.032 |

migraine were the limitations of our study. As Croall et al. [38] said, some changes in brain imaging before and after exposure to gluten are seen in celiac patients. Therefore, performing a brain imaging study in celiac patients with migraine before and after the start of a gluten-free diet can be useful in better understanding the physiopathology of migraine in these patients.

## Conclusion

Our study suggests that the prevalence of headache, especially migraine, in celiac patients is higher than healthy controls. In addition, abdominal pain, diarrhea, and constipation are more common in celiac patients with migraines than in celiac patients with non-migrainous headaches. Therefore, evaluation of celiac disease in patients with migraine and these simultaneous GI symptoms seems reasonable. Further interventional studies should also be performed on the role of a gluten-free diet in relieving headaches in celiac disease and other gluten-sensitive disorders.

## Supporting information

**S1 Checklist. Checklist for data collection.**
(PDF)

**S1 Text. Diagnostic criteria for migraine without aura.**
(PDF)

**S2 Text. Diagnostic criteria for migraine with typical aura.**
(PDF)

## Author Contributions

**Conceptualization:** Mohammad M. Fanaeian, Nazanin Alibeik, Azita Ganji, Golnaz Ekhlasi, Bijan Shahbazkhani.

**Data curation:** Mohammad M. Fanaeian, Nazanin Alibeik, Azita Ganji, Golnaz Ekhlasi, Bijan Shahbazkhani.

**Formal analysis:** Mohammad M. Fanaeian, Nazanin Alibeik, Hafez Fakheri, Golnaz Ekhlasi, Bijan Shahbazkhani.

**Investigation:** Nazanin Alibeik, Azita Ganji, Hafez Fakheri, Golnaz Ekhlasi.

**Methodology:** Nazanin Alibeik, Bijan Shahbazkhani.

**Project administration:** Nazanin Alibeik, Azita Ganji.

**Resources:** Nazanin Alibeik, Golnaz Ekhlasi.

**Software:** Nazanin Alibeik, Azita Ganji, Hafez Fakheri.

**Supervision:** Nazanin Alibeik, Bijan Shahbazkhani.

**Validation:** Nazanin Alibeik, Bijan Shahbazkhani.

**Visualization:** Nazanin Alibeik, Azita Ganji.

**Writing – original draft:** Nazanin Alibeik, Hafez Fakheri.

**Writing – review & editing:** Nazanin Alibeik, Bijan Shahbazkhani.

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
