## [Decision Letter · Decision Letter 0]

18 Mar 2021

PONE-D-21-03892

Prevalence of Migraine in Adults with Celiac Disease: A Case Control Cross-Sectional Study

PLOS ONE

Dear Dr. Alibeik,

Thank you for submitting your manuscript to PLOS ONE. After careful consideration, we feel that it has merit but does not fully meet PLOS ONE’s publication criteria as it currently stands. Therefore, we invite you to submit a revised version of the manuscript that addresses the points raised during the review process.

We look forward to receiving your revised manuscript.

Kind regards,

Tai-Heng Chen, M.D.

Academic Editor

PLOS ONE

Journal Requirements:

2) Please provide additional details regarding participant consent. In the ethics statement in the Methods and online submission information, please ensure that you have specified:

 - whether consent was obtained

 - whether consent was informed

 - what type of consent you obtained (for instance, written or verbal, and if verbal, how it was documented and witnessed).

 - if your study included minors, state whether you obtained consent from parents or guardians.

 - if the need for consent was waived by the ethics committee, please include this information.

3) Please include additional information regarding the checklist used in the study and ensure that you have provided sufficient details that others could replicate the analyses. For instance, if you developed the checklist as part of this study and it is not under a copyright more restrictive than CC-BY, please include a copy, in both the original language and English, as Supporting Information. If the checklist is published, please provide a citation to it.

4) Please revise Table 1 to replace p-values of "0.000" to "<0.001".

5) Please list the medical centers that participated in the study.

6)  We note that you have indicated that data from this study are available upon request. PLOS only allows data to be available upon request if there are legal or ethical restrictions on sharing data publicly. For information on unacceptable data access restrictions, please see http://journals.plos.org/plosone/s/data-availability#loc-unacceptable-data-access-restrictions.

Reviewers' comments:

Reviewer's Responses to Questions

**Comments to the Author**

1. Is the manuscript technically sound, and do the data support the conclusions?

Reviewer #1: Yes

Reviewer #2: Partly

Reviewer #3: Partly

2. Has the statistical analysis been performed appropriately and rigorously? 

Reviewer #1: Yes

Reviewer #2: I Don't Know

Reviewer #3: Yes

3. Have the authors made all data underlying the findings in their manuscript fully available?

Reviewer #1: Yes

Reviewer #2: Yes

Reviewer #3: Yes

4. Is the manuscript presented in an intelligible fashion and written in standard English?

Reviewer #1: Yes

Reviewer #2: No

Reviewer #3: Yes

5. Review Comments to the Author

Reviewer #1: 1)

Gluten-related disorders is the term for the diseases triggered by gluten, including celiac disease (CD), non-celiac gluten sensitivity (NCGS), gluten ataxia, dermatitis herpetiformis (DH) and wheat allergy.

Celiac diseae (CD) is one of the most common chronic, immune-mediated disorders, triggered by the eating of gluten.

The classic symptoms of untreated coeliac disease include pale, loose, or greasy stools (steatorrhoea), and weight loss or failure to gain weight. Other common symptoms may be subtle or primarily occur in organs other than the bowel itself.

Coeliac disease has been linked with a number of conditions. In many cases, it is unclear whether the gluten-induced bowel disease is a causative factor or whether these conditions share a common predisposition.

Headache is not a highly reported symptoms among the patients with this disease.

People of African, Japanese and Chinese descent are rarely diagnosed this disease.

That reflects a much lower prevalence of the genetic risk factors, such as HLA-B8.

Q1

Dose authors happen to have any data of ethnic background of this study?

Q2

I could not find this paper at Reference of this paper.

Could you please consider adding this paper to Reference of this study?

Headache. 2013 Feb;53(2):344-55.

Prevalence of migraine in patients with celiac disease and inflammatory bowel disease

Alexandra K Dimitrova 1, Ryan C Ungaro, Benjamin Lebwohl, Suzanne K Lewis, Christina A Tennyson, Mark W Green, Mark W Babyatsky, Peter H Green

Reviewer #2: This is a typical study to assess the prevalence of other comorbid disorders in headache and migraine patients. The authors compared Iranian celiac disease(CD) subjects with the control group to evaluate the prevalence of headache and migraine as well as the prevalence of some specific migrainous symptoms. The results are meaningful but not enough to support the conclusion. There are some points I concern about.

Line 35-36 the sentence should be revised. Headache prevalence is higher in CD patients but that does not mean the presence of headache could be a predictor of CD. You do not have a study to prove that the prevalence of celiac disease is higher in headache or migraine patients. Since CD itself is not a highly prevalent disease, we seldom take it into consideration when dealing with headache patients.

Line 40 change “disorder” to “disorders.”

Line 55-56 this statement seems not fully supported by this study. You do not have data to show a higher prevalence of CD in migraine patients. It may be possibly yes to migraine with concomitant GI symptoms.

Line 71-72 you’d better add “disabling” to describe the migraine symptoms which is also a characteristic of migraine.

Line 75 “but” is not necessary here.

Line 84-86 similar to Line 35-36, this description should be revised as it implies that migraine is a part of CD. Migraine itself is another disease entity, not a symptom. CD is just comorbidity.

Line 100 change “disorder” to “disorders.”

Line 103 “whom” should be “who”; “staffs” should be “staff.”

Line 106 “(P value =0.000)” means that the sex ratios of the study and control groups are not similar, right? It seems no need to emphasize.

Line 118 you just used a self-reported checklist to make the diagnosis of headache disorders. Why didn’t you use some validated screener such as the ID Migraine test?

Line 142 “Diabetes” should be “diabetes”. No need to capitalize.

Line 171-172 “headache” should be removed. “Bilateral” and “pulsatile” are characteristics of headache whereas aura, nausea, vomiting, photo- and phonophobia are associated symptoms. Also, about two-thirds of migraine headaches are unilateral which is one of the diagnostic criteria of migraine. “Bilateral” is not a characteristic of migraine.

Line 176 “;” should be “.” The following is another sentence.

Line 178 “Table 5” should be “Table 6.”

Line 179-180 Ruggieri et al. did the study of gluten sensitivity in children rather than CD in adults. Can you just compare the figures from different studies and make this conclusion?

Line 182 “be concluded” should be “conclude.”

Line 184-186 unclear to me. Please reword.

Line 193-194 as in Line 82-84 please reword. At least “refractory migraine” should be removed.

Table 4. please change “bilateral” to “unilateral” and correct the figures because the former is not a characteristic of migraine.

Line 228 please remove “headache” as some of them in the left column of table 4 are not characteristics of the headache but associated symptoms.

I suggest an English-editing service to correct some grammar mistakes.

Reviewer #3: Comment 1: The sex distribution were differed ( p= 0.000) between celiac and control groups (Table 1), this should be one of the limitations of this study.

Comment 2: Line 117 " Patients who reported a history of headaches..."; Line 132 " reported chronic headache..." . These two descriptions were inconsistent and should be clarified. "Headache " should be clearly defined, do you mean primary headache only, or secondary headache also included? "Chronic headache" is an ambiguous term, I am not able to know the meaning ( chronic primary headache history? or chronic daily headache? or others?) .

Comment 3: The authors didn't let us known how to perform the ICHD-3 based checklist (line 118) during questioning the patients, is it single-blinded or double-blinded?

Comment 4: According to Table 3, the both groups with more bilateral characters ( 67.2% and 65.2%) , this finding is different from the characters of general migraine patients, which "unilateral" were much more than "bilateral". This point should be clearly explained.

Comment 5: Table 5 revealed the two groups were differed in type I DM, abdominal pain, constipation and diarrhea, the possible causes of these differences needs to be explained.

Comment 6:Table 5 only compared CD subjects with migraine and non-migraine headache, however, it would be more interesting to compare the "symptoms and associated conditions" between CD with migraine headache and controls with migraine headache. If the authors have the data, please analyze and show in results and discussing them.

Comment 7: Based on comment 6, and the other results in this study, it's better to address the different characters between CD with migraine and migraine without CD (controls with migraine), and then provide possible explanations for them, in this way, this article may be more valuable.

6. PLOS authors have the option to publish the peer review history of their article (what does this mean?). If published, this will include your full peer review and any attached files.

Reviewer #1: **Yes: **Hikoaki Fukaura

Reviewer #2: No

Reviewer #3: **Yes: **Chi-Hsiang Chou

---

## [Author Response · Author response to Decision Letter 0]

27 Jun 2021

Responses to the reviewers:

Reviewr1: 

Q1: Dose authors happen to have any data of ethnic background of this study?

A1: No

Q2: Q2

I could not find this paper at Reference of this paper.

Could you please consider adding this paper to Reference of this study?

Headache. 2013 Feb;53(2):344-55.

Prevalence of migraine in patients with celiac disease and inflammatory bowel disease

Alexandra K Dimitrova 1, Ryan C Ungaro, Benjamin Lebwohl, Suzanne K Lewis, Christina A Tennyson, Mark W Green, Mark W Babyatsky, Peter H Green

A2: This reference No. is 22. 

Reviewer 2:

All items mentioned by the 2nd reviewer were considered. 

Reviewer 3:

Most items mentioned by the 3rd reviewer were considered. 

Table 4 was corrected; figures and percent are related to unilateral headaches. 

For some unknown reasons the prevalence of some diseases and symptoms are different in CD with and without migraine.

---

## [Decision Letter · Decision Letter 1]

11 Jul 2021

PONE-D-21-03892R1

Prevalence of Migraine in Adults with Celiac Disease: A Case Control Cross-Sectional Study

PLOS ONE

Dear Dr. Alibeik,

Thank you for submitting your manuscript to PLOS ONE. After careful consideration, we feel that it has merit but does not fully meet PLOS ONE’s publication criteria as it currently stands. Therefore, we invite you to submit a revised version of the manuscript that addresses the points raised during the review process.

We look forward to receiving your revised manuscript.

Kind regards,

Tai-Heng Chen, M.D.

Academic Editor

PLOS ONE

Reviewers' comments:

Reviewer's Responses to Questions

**Comments to the Author**

1. If the authors have adequately addressed your comments raised in a previous round of review and you feel that this manuscript is now acceptable for publication, you may indicate that here to bypass the “Comments to the Author” section, enter your conflict of interest statement in the “Confidential to Editor” section, and submit your "Accept" recommendation.

Reviewer #2: All comments have been addressed

Reviewer #3: (No Response)

2. Is the manuscript technically sound, and do the data support the conclusions?

Reviewer #2: Yes

Reviewer #3: Partly

3. Has the statistical analysis been performed appropriately and rigorously? 

Reviewer #2: I Don't Know

Reviewer #3: I Don't Know

4. Have the authors made all data underlying the findings in their manuscript fully available?

Reviewer #2: Yes

Reviewer #3: Yes

5. Is the manuscript presented in an intelligible fashion and written in standard English?

Reviewer #2: Yes

Reviewer #3: Yes

6. Review Comments to the Author

Reviewer #2: The article is much better after revising. But the authors used the terms ‘migraine headache’, and ‘migraine’ interchangeably. 'Migraine headache' is not a good term as it may confuse the readers. Migraine is a specific headache disorder, no need to add 'headache' after it. If you want to describe the symptom ‘headache’, please use 'migrainous headache'. The 'migrainous headache' is a specific headache that presents unilateral, pulsating and disabling characteristics. Please recheck the whole article for this problem.

Line 39 As mentioned above, if you mean the symptom ‘headache’, you should use 'migrainous headache'. If you mean the disease ‘migraine’, you may say 'to evaluate … the comorbidity of migraine in CD…'.

Line 57 The conclusion should be revised again. The migraine itself would be associated with some GI symptoms by ICHD definition. If following this conclusion, every migraine patient should be evaluated for CD. You have to specify what GI symptoms could be related to CD.

Line 217 I don’t mean ‘ be conclude’, but just ‘conclude’. If you want to keep ‘be concluded’, it is okay.

Line 229 As mentioned above, please revise the conclusion.

Reviewer #3: The authors didn't reply my comments in a formal type. Due to the problem, I can't realize my comments were adequately answered or not.The authors have to answer my previous comments one by one.

7. PLOS authors have the option to publish the peer review history of their article (what does this mean?). If published, this will include your full peer review and any attached files.

Reviewer #2: **Yes: **Tzu-Chou Huang

Reviewer #3: No

---

## [Author Response · Author response to Decision Letter 1]

2 Aug 2021

Responses to the Reviewers:

Reviewr1: 

Q1: Dose authors happen to have any data of ethnic background of this study?

A1: No

Q2: I could not find this paper at Reference of this paper.

Could you please consider adding this paper to Reference of this study?

Headache. 2013 Feb;53(2):344-55.

Prevalence of migraine in patients with celiac disease and inflammatory bowel disease

Alexandra K Dimitrova 1, Ryan C Ungaro, Benjamin Lebwohl, Suzanne K Lewis, Christina A Tennyson, Mark W Green, Mark W Babyatsky, Peter H Green

A2: This reference No. is 22. 

Reviewer 2:

As you mentioned, the terms “migraine headache” and “non-migraine headache” were replaced by the “migraine” and “non- migrainous headache” throughout the manuscript.

Conclusion was revised in “abstract” and “conclusion” sections (Lines 58-60 and Lines 268-271)

Reviewer 3:

Comment 1: As you mentioned, the sex distribution was differed (P= 0.000) between celiac and control groups, and that was one of the limitations of this study and we mentioned that in discussion (line 258)

Comment 2: As you said, the term “chronic headache” was replaced. We mentioned in the method section. (Line 122-125)

Comment 3: The ICHD-3 based checklist was written in Supporting information caption and it was filled single-blinded.

Comment 4: Table 4 was revised; figures and percent are related to unilateral headaches.

Comment 5: GI symptoms and migraine were explained (lines 238-243, discussion section). Migraine and type 1 were explained. (Lines 86-90, Introduction section/ lines 244-247, discussion section)

Comment 6 & 7: Thanks for your good comment. Unfortunately, this data is not available in the control group.

---

## [Decision Letter · Decision Letter 2]

11 Aug 2021

PONE-D-21-03892R2

Prevalence of Migraine in Adults with Celiac Disease: A Case Control Cross-Sectional Study

PLOS ONE

Dear Dr. Alibeik,

Thank you for submitting your manuscript to PLOS ONE. After careful consideration, we feel that it has merit but does not fully meet PLOS ONE’s publication criteria as it currently stands. Therefore, we invite you to submit a revised version of the manuscript that addresses the points raised during the review process.

We look forward to receiving your revised manuscript.

Kind regards,

Tai-Heng Chen, M.D.

Academic Editor

PLOS ONE

Journal Requirements:

Reviewers' comments:

Reviewer's Responses to Questions

**Comments to the Author**

1. If the authors have adequately addressed your comments raised in a previous round of review and you feel that this manuscript is now acceptable for publication, you may indicate that here to bypass the “Comments to the Author” section, enter your conflict of interest statement in the “Confidential to Editor” section, and submit your "Accept" recommendation.

Reviewer #2: All comments have been addressed

Reviewer #3: All comments have been addressed

2. Is the manuscript technically sound, and do the data support the conclusions?

Reviewer #2: Yes

Reviewer #3: Yes

3. Has the statistical analysis been performed appropriately and rigorously? 

Reviewer #2: I Don't Know

Reviewer #3: Yes

4. Have the authors made all data underlying the findings in their manuscript fully available?

Reviewer #2: Yes

Reviewer #3: Yes

5. Is the manuscript presented in an intelligible fashion and written in standard English?

Reviewer #2: Yes

Reviewer #3: Yes

6. Review Comments to the Author

Reviewer #2: Much better. But there are still two points.

Line 159. The term ‘primary headache’ is a specific term that includes migraine, tension-type headache, trigeminal autonomic cephalalgia, and others only. I believe you mean just a complaint of headache here. Maybe you can remove the word ‘primary’.

Line 240. Common GI symptom of migraine is nausea/vomiting, not abdominal pain. According to ICHD-3 (your reference 34), there is no abdominal pain in the description of migraine. You’d better change it to ‘nausea/vomiting’.

Reviewer #3: About my previous comment 4, I think the table 4 should divide nausea, vomiting, photophobia, phonophobia into four different lines and calculate the individual percentages and P-values. In this way, the data would be more clarified and more easily to be realized.

7. PLOS authors have the option to publish the peer review history of their article (what does this mean?). If published, this will include your full peer review and any attached files.

Reviewer #2: **Yes: **Tzuchou Huang

Reviewer #3: No

---

## [Author Response · Author response to Decision Letter 2]

5 Sep 2021

Responses to the Reviewers:

Reviewer 2:

As you mentioned, the term “primary headache” was replaced by the “headache” in manuscript.

According to ICHD-3, abdominal pain can be one of the episodic syndromes associated with migraine. (The International Classification of Headache Disorders, 3rd edition page 653)

Reviewer 3:

In data collection, nausea and vomiting were both considered as a single manifestation and were not considered individually.

In data collection, photophobia and phonophobia were both considered as a single manifestation and were not considered individually.

---

## [Decision Letter · Decision Letter 3]

22 Sep 2021

PONE-D-21-03892R3Prevalence of Migraine in Adults with Celiac Disease: A Case Control Cross-Sectional StudyPLOS ONE

Dear Dr. Alibeik,

Thank you for submitting your manuscript to PLOS ONE. After careful consideration, we feel that it has merit but does not fully meet PLOS ONE’s publication criteria as it currently stands. Therefore, we invite you to submit a revised version of the manuscript that addresses the points raised during the review process.

We look forward to receiving your revised manuscript.

Kind regards,

Tai-Heng Chen, M.D.

Academic Editor

PLOS ONE

Journal Requirements:

Reviewers' comments:

Reviewer's Responses to Questions

**Comments to the Author**

1. If the authors have adequately addressed your comments raised in a previous round of review and you feel that this manuscript is now acceptable for publication, you may indicate that here to bypass the “Comments to the Author” section, enter your conflict of interest statement in the “Confidential to Editor” section, and submit your "Accept" recommendation.

Reviewer #3: All comments have been addressed

2. Is the manuscript technically sound, and do the data support the conclusions?

Reviewer #3: Yes

3. Has the statistical analysis been performed appropriately and rigorously? 

Reviewer #3: Yes

4. Have the authors made all data underlying the findings in their manuscript fully available?

Reviewer #3: Yes

5. Is the manuscript presented in an intelligible fashion and written in standard English?

Reviewer #3: Yes

6. Review Comments to the Author

Reviewer #3: About my comment 4, and ICHD 3, (the diagnostic criteria of migraine without aura, the criteria D: During headache at least one of the following: 1. nausea and/or vomiting; 2.photophobia and phonophobia ) I think in the table 4, the row " photophobia and/or phonophobia " should be corrected into "photophobia and phonophobia". Due to the change, I consider the data in this part needs to be calculated again.

7. PLOS authors have the option to publish the peer review history of their article (what does this mean?). If published, this will include your full peer review and any attached files.

Reviewer #3: No

---

## [Author Response · Author response to Decision Letter 3]

6 Oct 2021

Responses to the Reviewers:

Reviewer 3:

Thanks for your comment, as you mentioned, based on the ICHD 3, (the diagnostic criteria of migraine without aura, the criteria D) in data collection, photophobia and phonophobia were both considered as a single manifestation and were not considered individually.

 In Table 4, the term "photophobia and phonophobia" is correct and the word "or" was incorrectly mentioned, which was corrected.

---

## [Decision Letter · Decision Letter 4]

21 Oct 2021

Prevalence of Migraine in Adults with Celiac Disease: A Case Control Cross-Sectional Study

PONE-D-21-03892R4

Dear Dr. Alibeik,

We’re pleased to inform you that your manuscript has been judged scientifically suitable for publication and will be formally accepted for publication once it meets all outstanding technical requirements.

Kind regards,

Tai-Heng Chen, M.D.

Academic Editor

PLOS ONE

Reviewers' comments:

Reviewer's Responses to Questions

**Comments to the Author**

1. If the authors have adequately addressed your comments raised in a previous round of review and you feel that this manuscript is now acceptable for publication, you may indicate that here to bypass the “Comments to the Author” section, enter your conflict of interest statement in the “Confidential to Editor” section, and submit your "Accept" recommendation.

Reviewer #3: (No Response)

2. Is the manuscript technically sound, and do the data support the conclusions?

Reviewer #3: Yes

3. Has the statistical analysis been performed appropriately and rigorously? 

Reviewer #3: I Don't Know

4. Have the authors made all data underlying the findings in their manuscript fully available?

Reviewer #3: Yes

5. Is the manuscript presented in an intelligible fashion and written in standard English?

Reviewer #3: Yes

6. Review Comments to the Author

Reviewer #3: (No Response)

7. PLOS authors have the option to publish the peer review history of their article (what does this mean?). If published, this will include your full peer review and any attached files.

Reviewer #3: No

---

## [Editor Report · Acceptance letter]

25 Oct 2021

PONE-D-21-03892R4 

Prevalence of Migraine in Adults with Celiac Disease: A Case Control Cross-Sectional Study 

Dear Dr. Alibeik:

I'm pleased to inform you that your manuscript has been deemed suitable for publication in PLOS ONE. Congratulations! Your manuscript is now with our production department. 

Kind regards, 

on behalf of

Dr. Tai-Heng Chen 

Academic Editor

PLOS ONE